# Gender Differences in the Pharmacological Actions of Pegylated Glucagon-Like Peptide-1 on Endothelial Progenitor Cells and Angiogenic Precursor Cells in a Combination of Metabolic Disorders and Lung Emphysema

**DOI:** 10.3390/ijms20215414

**Published:** 2019-10-30

**Authors:** Olga Victorovna Pershina, Angelina Vladimirovna Pakhomova, Darius Widera, Natalia Nicolaevna Ermakova, Anton Alexandrovich Epanchintsev, Edgar Sergeevich Pan, Vyacheslav Andreevich Krupin, Olga Evgenevna Vaizova, Olesia Dmitrievna Putrova, Lubov Alexandrovna Sandrikina, Irina Vitalevna Kurochkina, Sergey Georgievich Morozov, Aslan Amirkhanovich Kubatiev, Alexander Mikhaylovich Dygai, Evgenii Germanovich Skurikhin

**Affiliations:** 1Laboratory of Regenerative Pharmacology, Goldberg ED Research Institute of Pharmacology and Regenerative Medicine, Tomsk National Research Medical Centre of the Russian Academy of Sciences, 634028 Tomsk, Russia; angelinapakhomova2011@gmail.com (A.V.P.); nejela@mail.ru (N.N.E.); artifexpan@gmail.com (E.S.P.); vakrupin88@gmail.com (V.A.K.); olesya.putrova@mail.ru (O.D.P.); ermolaeva_la@mail.ru (L.A.S.); irinakuro4kina93@yandex.ru (I.V.K.); amdygay@gmail.com (A.M.D.); eskurihin@inbox.ru (E.G.S.); 2Stem Cell Biology and Regenerative Medicine Group, School of Pharmacy, University of Reading, Whiteknights campus, Reading RG6 6AP, UK; d.widera@reading.ac.uk; 3Branch Federal State Unitary Enterprise “Scientific and Production Association for Immunological Preparations “Microgen” of Ministry of Health of the Russian Federation “SIC “Virion” in Tomsk, 634040 Tomsk, Russia; a.a.epanchintsev@gmail.com; 4Department of Pharmacology, Siberian State Medical University, 634050 Tomsk, Russia; vaizova@mail.ru; 5Institute of General Pathology and Pathophysiology, 125315 Moscow, Russia; biopharm@list.ru (S.G.M.); niiopp@mail.ru (A.A.K.)

**Keywords:** gender differences, dyslipidemia, obesity, hyperglycemia, pulmonary emphysema, endothelial progenitor cells, angiogenic precursor cells, pegylated glucagon-like peptide 1, and endothelial regeneration

## Abstract

In clinical practice, the metabolic syndrome (MetS) is often associated with chronic obstructive pulmonary disease (COPD). Although gender differences in MetS are well documented, little is known about sex-specific differences in the pathogenesis of COPD, especially when combined with MetS. Consequently, it is not clear whether the same treatment regime has comparable efficacy in men and women diagnosed with MetS and COPD. In the present study, using sodium glutamate, lipopolysaccharide, and cigarette smoke extract, we simulated lipid metabolism disorders, obesity, hyperglycemia, and pulmonary emphysema (comorbidity) in male and female C57BL/6 mice. We assessed the gender-specific impact of lipid metabolism disorders and pulmonary emphysema on angiogenic precursor cells (endothelial progenitor cells (EPC), pericytes, vascular smooth muscle cells, cells of the lumen of the nascent vessel), as well as the biological effects of pegylated glucagon-like peptide 1 (pegGLP-1) in this experimental paradigm. Simulation of MetS/COPD comorbidity caused an accumulation of EPC (CD45^−^CD31^+^CD34^+^), pericytes, and vascular smooth muscle cells in the lungs of female mice. In contrast, the number of cells involved in the angiogenesis decreased in the lungs of male animals. PegGLP-1 had a positive effect on lipids and area under the curve (AUC), obesity, and prevented the development of pulmonary emphysema. The severity of these effects was stronger in males than in females. Furthermore, PegGLP-1 stimulated regeneration of pulmonary endothelium. At the same time, PegGLP-1 administration caused a mobilization of EPC (CD45^−^CD31^+^CD34^+^) into the bloodstream in females and migration of precursors of angiogenesis and vascular smooth muscle cells to the lungs in male animals. Gender differences in stimulatory action of pegGLP-1 on CD31^+^ endothelial lung cells in vitro were not observed. Based on these findings, we postulated that the cellular mechanism of in vivo regeneration of lung epithelium was at least partly gender-specific. Thus, we concluded that a pegGLP-1-based treatment regime for metabolic disorder and COPD should be further developed primarily for male patients.

## 1. Introduction

In clinical practice, metabolic syndrome (MetS) is defined as the presence of at least three of the following five conditions: abdominal obesity, elevated triglyceride levels, low serum high-density lipoprotein (HDL), high blood pressure (hypertension), and elevated levels of blood glucose [1].

Gender differences in glucose homeostasis and energy balance are important factors in the development of MetS in men and women [2]. In general, women are at greater risk of metabolic disorders due to a higher fat/muscle mass ratio and more subcutaneous fat tissue. Consequently, females have higher insulin resistance than men. This is associated with hormonal differences and diet [3,4,5,6]. Gender-specific differences in the lipoprotein profile are well described [7]. In addition, gender-specific mechanisms beyond the scope of this study contribute to a higher accumulation of adipose tissue in females, whereas mobilization of fat reserves tends to be more effective in men.

In addition to MetS, gender-specific differences in major metabolic pathways have been suggested to contribute to the pathogenesis of various lung diseases [8,9].

Clinically, MetS has been found to be more frequent in patients with chronic obstructive pulmonary disease (COPD). This patient cohort has a higher body mass index compared to patients with COPD alone [10]. The clinical picture and course of COPD often differ in women and men [11]. Importantly, COPD with MetS is more frequent in female patients with a prevalence of 18.5% in men and 38.5% in women [12].

Fatty acids are an important source of energy. In this context, catabolism of fatty acids yields more Adenosine triphosphate (ATP) per mole than in the oxidation of glucose [13,14,15].

Recent reports indicate that changes in lipid metabolism allow lung tissue to meet the energy needs of patients with COPD [16]. On the other hand, changes in lipid metabolism (catabolism, anabolism) contribute to the pathogenesis of COPD. In particular, obesity initiates anabolic pathways involved in the synthesis of pro-inflammatory molecules [16]. This mechanism reduces lung function and increases the severity of COPD [17,18].

Understanding the gender-specific mechanisms of a disease is important for finding individual and personalized treatment strategies in patients with MetS and COPD.

In a recent review, Kim J.H. and colleagues highlighted gender-specific differences in the histology and functional activity of the lungs in patients with COPD and MetS [12]. Briefly, the authors reported that in comparison with men, women have fewer alveoli. Furthermore, their airway diameter is relatively small compared to the size of the lungs. Thus, lung function tends to decline sharply as obesity increases. Moreover, estrogens can reduce lung function. In 2018, Zore T. et al. drew attention to adipocyte progenitor cells as a potential factor affecting sex differences in adipose tissue enlargement [19]. In contrast, they postulated that hematopoietic stem cells are a factor contributing to gender differences in the inflammatory response.

Previously, we demonstrated that incretin glucagon-like peptide-1 (GLP-1) administration could stimulate the regeneration of lung endothelium in lung emphysema in obese mice [20]. The regenerative effects of GLP-1 have been at least partly mediated by effects on endothelial progenitor cells. However, in vivo, GLP-1 is rapidly metabolized by dipeptidyl peptidase-4, leading to a relatively low regenerative activity of the hormone. It is well described that pegylation can be used to preserve pharmacologically active molecules.

The aim of this research was to study gender differences in the treatment of hyperglycemia, dyslipidemia and obesity, emphysema, and alveolar endothelial injury with pegylated GLP-1 (pegGLP-1) in a C57BL/6 mice model of obesity and emphysema. We investigated the effect of pegGLP-1 on bone marrow, circulating in the blood, and pulmonary endothelial progenitor cells and other cells involved in angiogenesis in female and male animals to assess potential gender-specific differences.

## 2. Results

### 2.1. The Effect of GLP-1 and pegGLP-1 on Lee Index and Body Mass Index

To confirm obesity in the male and female C57BL/6 mice on p189—mice that received MSG (monosodium glutamate)—Lee and body mass index (BMI) indexes were assessed. Simulations of pulmonary emphysema did not affect the Lee index and BMI in animals of f3 and m3 groups (mice with lung emphysema) compared to intact controls (Figure 1). In the modeling of metabolic disorders (obesity and hyperglycemia), and in the modeling of metabolic disorders (obesity and hyperglycemia) and lungs emphysema, we observed an increase in the Lee and BMI indexes in females (groups f2 (mice with metabolic disorders) and f4 (mice with metabolic disorders and lung emphysema)) and males (groups m2 and m4), while in males, the increase in parameters was more pronounced compared to females.

GLP-1 or pegGLP-1 treatment had no effect on the Lee index of females and males in metabolic disorders (obesity and hyperglycemia) and emphysema compared with untreated mice of groups f4 and m4 (Figure 1b). Meanwhile, drugs significantly reduced BMI in females of groups f5 (mice with metabolic disorders and lung emphysema treated with GLP-1) and f6 (mice with metabolic disorders and lung emphysema treated with peg-GLP-1), and males of groups m5 and m6. The therapeutic effect in males m6 was more pronounced compared to females f6. This section may be divided by subheadings. It should provide a concise and precise description of the experimental results, their interpretation, as well as the experimental conclusions that can be drawn.

### 2.2. Changes in Serum Lipid Parameters in Emphysema, Metabolic Disorders, and the Combination of Metabolic Disorders and Emphysema

Dyslipidemia is a key component of metabolic disorders (MD) and often occurs with obesity. We studied levels of cholesterol, triglycerides (TG), high-density lipoprotein (HDL), low-density lipoprotein (LDL), and very-low-density lipoprotein (VLDL) in the serum of male and female C57BL/6 mice on p189. The m2 group showed a more pronounced increase in cholesterol, TG, HDL, and VLDL compared with the f2 group. In contrast, in group f2, there was a more marked increase in LDL than in group m2 (Figure 2c). We also observed gender-dependent differences in serum lipid levels in the development of emphysema. Thus, the levels of TG and LDL in the m3 group increased, while in the f3 group, these indicators decreased (Figure 2a,c). It should be noted that the levels of cholesterol, LDL, and HDL in males and females with emphysema of the lungs changed the same type—they increased.

The combination of MD and lung emphysema revealed differences in the lipid profile of males and females. In female mice of group f4 (mice with MD and lung emphysema), serum LDL concentrations increased compared to f1 (intact control) and f2 (Figure 2c). In the m4 group, this figure did not change significantly. Decrease in triglycerides (TG) and VLDL and increase in cholesterol, VLDL in the blood serum was observed in mice m4 and f4, but the degree of severity of changes was more pronounced in males.

### 2.3. The Effect of GLP-1 and pegGLP-1 on Lipid Parameters of Blood Serum with a Combination of Metabolic Disorders and Emphysema

On p189, we studied the effectiveness of GLP-1 and pegGLP-1 on fat metabolism in mice of different sexes under the conditions of MSG and cigarette smoke extract (CSE) administration (a combination of MD and lung emphysema).

The introduction of GLP-1 to m5 males (with an MD and lung emphysema) reduced the concentration of TG (1.7 times) and VLDL (1.7 times) compared to m4 males. In contrast, the concentration of cholesterol and LDL increased (Figure 2c). At the same time, GLP-1 increased cholesterol and HDL cholesterol levels in females of the f5 group (with MD and lung emphysema) and did not affect other parameters.

The introduction of pegGLP-1 to m6 (with MD and lung emphysema) caused an increase in cholesterol, HDL, and LDL levels compared to the m4 group (Figure 2). On the contrary, the level of TG decreased (by 1.5 times) in group f6 females—affected by pegGLP-1—compared to group f4. The level of VLDL was reduced, while the level of LDL increased.

In addition, we studied gender differences in the ratio of triglycerides to high-density lipoproteins (TG/HDL). As can be seen from Figure 2e, the parameter value in females of the f1 group was significantly higher compared to males of the m1 group (6.8 times). Modeling of metabolic disorders, emphysema of the lungs, and a combination of metabolic disorders and emphysema of the lungs caused an increase in TG/HDL in males of group m2 (3 times), m3 (2.8 times) and m4 (1.6 times), respectively, compared to intact control. In females of group f2, the parameter increased slightly (by 8%), in groups f3 and f4, on the contrary, we observed its decrease by 69% and 57.4%, respectively, compared to f1. The GLP-1 treatment helped to reduce the ratio of TG/HDL in males and females in a combination of MD and lung emphysema, while the effect of the drug was most pronounced in the m5 group. PegGLP-1 did not affect the studied parameter in males in the m6 group and reduced it in females in the f6 group by 35%.

### 2.4. GLP-1 and PegGLP-1 Effect on Area Under the Curve (AUC) During the Glucose Tolerance Test

The day before the removal of animals from the experiment (p188), the glucose tolerance test (GTT) and the calculation of the area under the curve for blood glucose (AUC) were conducted. After the introduction of MSG, there was a natural increase in AUC in males of the m2 group (two times), and it was significantly less in females of the f2 group (19%) compared to the control groups (Figure 3). Interestingly, the introduction of cigarette smoke extract (CSE) also increased AUC in males of the m3 group (by 21%), while the parameter did not change in the f3 group. Simulations of the combined pathology did not affect AUC in f4 females, but in m4 males, the parameter increased 1.8 times compared to the m1 group.

The GLP-1 or pegGLP-1 treatment decreased AUC in male mice of m5 (13%) and m6 (18%), respectively (Figure 3). PegGLP-1 (11%) was more effective in females with comorbidity, but not GLP-1 (5%).

### 2.5. Morphological Study of Lung

Lung injuries caused by lipopolysaccharide (LPS) and CSE were similar in group f3 females and group m3 male mice. Thus, by the p148 (24 h before treatment), the animals developed moderately diffused lungs emphysema. In addition, histological preparations of the lungs revealed an increase in the size of the alveoli and alveolar passages and single ruptures of the alveolar septa due to damage to elastic membranes. In mice of groups (m3 and f3), pulmonary hyperemia and diapedesis of erythrocytes in the lumen of the alveoli were revealed; the walls of the alveoli were thickened due to inflammatory infiltration by macrophages (Figure 4b). At the same time, macrophages and single neutrophils were found in the lumen of the alveoli, and peribronchial lymph-macrophage infiltrates were observed. On p189, as well as on p148, emphysema of the lungs in group f3 and group m3 mice was diffused. Meanwhile, on p189, the thinning of the walls of the alveoli, the number of ruptures and atelectasis of the pulmonary tissue, and the area of emphysema in these mice were more significant compared to p148 (Figure 4b). The area of emphysema in the males was superior to that of the females.

When modeling metabolic disorders in the lungs of group f2 and group m2 animals, we found hyperemia of small and large vessels, a large number of macrophages, and single neutrophils on p189 (Figure 4). In males, these parameters were more pronounced than in females.

The appointment of MSG and CSE (a combination of MD and lung emphysema) caused the development of focal lung emphysema of moderate severity in males (group m4) and females (group f4) on p189 (Figure 4). The area of emphysema in females (group f4) was smaller than in males (group m4). In the lungs of mice treated with a combination of metabolic disorders and lung emphysema, we found groups of enlarged alveoli and alveolar passages, and ruptures of the alveolar walls. In the alveoli, there have been sporadic neutrophils and a greater number of macrophages compared to females f3 and males m3. It should be noted that the area of emphysema in group f4 and group m4 was inferior to that in the group f3 and group m3, respectively.

GLP-1 and pegGLP-1 treatment slightly reduced the inflammatory infiltration by macrophages of the lungs of females and males under the conditions of MSG and CSE administration on p189. In the lungs of females of groups f5 and f6, the area of emphysema-enlarged tissue significantly decreased compared to untreated females of group f4 (Figure 4). The pegGLP-1 effect was higher than that of GLP-1.

### 2.6. Immunohistochemical Lung Study

The CSE introduction significantly reduced the number of CD31-expressing cells in the pulmonary tissue of m3 and f3 mice compared to m1 and f1 mice by the p189 (Figure 5). With the MSG and CSE introduction (a combination of MD and lung emphysema), the reduction of the number of CD31^+^ cells in the lungs of mice (group m4 and f4) was more significant than in m3 and f3 mice. When modeling pathology, the decrease in CD31 expression in the lungs of males was more significant than in females.

GLP-1 and pegGLP-1 treatment caused a significant increase in the number of CD31^+^ cells in mice lungs under MSG and CSE administration compared to untreated mice with MD and emphysema of the lungs (Figure 5). In this case, the therapeutic effect of males was higher than that of females.

### 2.7. Study of Stem Antigens, Epithelial and Endothelial Cells, and Other Cells Using Flow Cytometric Analysis

#### 2.7.1. Lung

On p189, we studied the precursors and mature cells content in the lungs of healthy mice (males and females). In the lungs of the m1 male group, we found a significantly greater number of endothelial cells (CD45^−^CD31^+^CD34^+^ and CD31^+^CD34^+^CD146^+^) and the precursors of angiogenesis (CD45^−^CD309^+^CD117^+^), cells in the lumen of the nascent vessel (CD31^+^CD34^−^), than the females of group f1 (Figure 6). At the same time, the number of vascular smooth muscle cells (CD31^−^CD34^+^CD146^+^) in the m1 group was inferior to that in the f1 group, and no significant gender differences in the content of pericytes (CD31^−^CD34^−^CD146^+^) were revealed.

MSG introduction caused an increase in the number of EPC and precursors of angiogenesis, vascular smooth muscle cells, pericytes in females of group f2 (mice with MD) compared to females of group f1 (Figure 6). In males of the m2 group, we found a decrease in the number of EPC and angiogenesis precursors, nascent vessel lumen cells, with vascular smooth muscle cells and pericytes accumulating in the lungs.

The LPS and CSE introduction caused a significant increase in the number of cells of the lumen of the nascent vessel and precursors of angiogenesis in group f3 compared to group f1 (Figure 6). In contrast, in the lungs of males of the m3 group, we observed a significant decrease in the number of EPC (CD45^−^CD31^+^CD34^+^), vascular smooth muscle cells, and angiogenesis precursors compared to the m1 group. On the contrary, the number of pericytes increased.

Modeling of MD and lung emphysema by MSG and CSE in females of group f4 caused a significant increase in the number of EPC (CD45^−^CD31^+^CD34^+^), vascular smooth muscle cells, and pericytes compared to group f1 (Figure 6). In contrast, the combination of MD and emphysema significantly reduced the number of EPC (CD45^−^CD31^+^CD34^+^), precursors of angiogenesis, and pericytes in the m4 group compared to healthy males.

The GLP-1 treatment caused similar changes in mice of f5 and m5 groups, such as a decrease in the number of pulmonary EPC (CD45^−^CD31^+^CD34^+^), vascular smooth muscle cells, and an increase in the number of angiogenesis precursors (Figure 6). Inter-gender differences were in the additional reduction of the population of pulmonary EPC (CD31^+^CD34^+^CD146^+^) in the m5 group. In addition, GLP-1 increased the number of pericytes in the lungs of m5 mice and reduced their number in f5 mice compared to untreated animals.

The pegGLP-1 reduced the population of cells in the lumen of the nascent vessel but increased the number of vascular smooth muscle cells in group m6 (Figure 6). Changes in the cells of the lungs of group f6 females to the pegGLP-1 introduction were as follows: the number of precursors of angiogenesis, the pericytes, and vascular smooth muscle cells decreased in comparison with the female group f5, on the other hand, the number of EPC (CD45^−^CD31^+^CD34^+^) increased.

#### 2.7.2. Bone Marrow

On p189, we revealed gender differences in the populations of immature endothelial bone marrow cells of healthy mice. In m1 males, the number of EPC (CD31^+^CD34^+^CD146^+^) (5.5 times) and angiogenesis precursors (4.58 times) significantly exceeded that of f1 females (Figure 7).

The MSG introduction caused a decrease in the number of bone marrow EPC (CD31^+^CD34^+^CD146^+^) and angiogenesis precursors in m2 males. On the contrary, in MSG-treated f2 females, we observed a significant increase in the number of these cells (Figure 7d).

The LPS and CSE introduction caused a significant increase in the number of all endothelial cells studied in the bone marrow of females of group f3 compared to group f1 (Figure 7c,d). In turn, in m3 males, we observed a selective accumulation of EPC (CD45^−^CD31^+^CD34^+^) 11 times compared to healthy males.

Modeling of MD and lung emphysema caused an increase in the content of bone marrow EPC (CD31^+^CD34^+^CD146^+^) and angiogenesis precursors in f4 females compared to healthy females (Figure 7 d,e). In males of the m4 group, a combination of MD and pulmonary emphysema led to the accumulation of EPC (CD45^−^CD31^+^CD34^+^) and precursors of angiogenesis.

The GLP-1 introduction increased the number of EPC (CD45^−^CD31^+^CD34^+^) and reduced the number of CD31^+^CD34^+^CD146^+^-endothelial cells in the bone marrow of f5 and m5 mice compared to f4 and m4 mice, respectively (Figure 7). We found gender differences in the reaction of angiogenesis precursors to treatment: in females of group f5, the number of CD45^−^CD309^+^CD117^+^-cells was 64% higher than that in group f4; males of m5 showed a tendency to decrease the number of these cells.

The pegGLP-1 treatment increased the number of bone marrow CD45^−^CD31^+^CD34^+^ EPC and precursors of angiogenesis in mice of groups f6 and m6 compared to untreated mice in terms of MD and lung emphysema (Figure 7c). On the other hand, in these groups, we found a decrease in the number of CD31^+^CD34^+^CD146^+^-EPC, which was more pronounced in the m6 group than in the f5 group.

#### 2.7.3. Blood

We found gender differences in the blood cells of healthy animals. Thus, in the m1 group, the number of circulating CD45^−^CD31^+^CD34^+^-EPC (by 63%), CD31^+^CD34^+^CD146^+^-EPC (by 94%), and vascular smooth muscle cells (by 99.5%) was inferior to that in the f1 group (Figure 8). In contrast, the number of pericytes (21.5 times) and lumen cells of the nascent vessel (1.87 times) in healthy males was higher than in healthy females.

The MSG introduction caused a decrease in the number of CD45^−^CD31^+^CD34^+^-EPCs, nascent vessel lumen cells, and pericytes in the blood of females and males compared to the corresponding healthy mice (Figure 8). Gender differences included an increase in the number of CD31^+^CD34^+^CD146^+^-EPCs and vascular smooth muscle cells in females (f2) and a reduction in these cell populations in males (m2).

LPS and CSE introduction caused a significant increase in the number of EPC (CD45^−^CD31^+^CD34^+^ and CD31^+^CD34^+^CD146^+^), vascular smooth muscle cells, and pericytes in the blood of m3 males compared to healthy m1 males (Figure 8). Meanwhile, in the blood of f3 group, we observed a decrease in the number of endothelial cells (CD45^−^CD31^+^CD34^+^ and CD31^+^CD34^+^CD146^+^), vascular smooth muscle cells, and cells of the lumen of the nascent vessels; the only exception was the pericytes, the number of which sharply increased compared to group f1.

The reaction of circulating blood cells of females of the f4 group in the modeling of MD and lung emphysema coincided with that which was identified in females with emphysema (Figure 8). In males of the m4 group, a coincidence of the reaction (increase) of vascular smooth muscle cells and pericytes with that of males of the m3 group (emphysema of the lungs) was revealed. At the same time, the number of endothelial cells (CD45^−^CD31^+^CD34^+^ and CD31^+^CD34^+^CD146^+^) and nascent vessel cells in the blood of m4 males, on the contrary, decreased and was less than in the m1 group.

GLP-1 reduced the number of all studied cells in f5 females compared to untreated f4 females: endothelial cell populations (CD45^−^CD31^+^CD34^+^ and CD31^+^CD34^+^CD146^+^) and vascular smooth muscle cells were most significantly reduced (Figure 8). In contrast, GLP-1 increased the number of CD31^+^CD34^+^CD146^+^-endothelial cells, vascular smooth muscle cells, and blood pericytes in m5 males compared to m4 males.

PegGLP-1 treatment caused an increase in the number of endothelial cells and vascular smooth muscle cells in the blood of females and males f6 and m6 compared to untreated animals in conditions of MD and lung emphysema (Figure 8). Gender differences were expressed in the accumulation of nascent vessel lumen cells in f6 females, and the opposite reaction of pericytes: the number of these cells in f6 females decreased, and it increased in m6 males.

### 2.8. Study of GLP-1 and PegGLP-1 Effect on CD31^+^ Lung Cells In Vitro

The effects of GLP-1 and pegGLP-1 on some parameters of CD31^+^ cells obtained from the lungs of females of f1 and f4 groups and males of m1 and m4 groups were studied in vitro. GLP-1 or pegGLP-1 were introduced into the cell culture enriched with CD31^+^ cells, and the final concentration of these drugs in the culture was 10^−7^ M.

Figure 9d shows that affected by GLP-1 and pegGLP-1, the number of apoptotic CD31^+^ cells in the f1 and m1 groups did not change. Meanwhile, in the f4 and m4 groups, GLP-1 and pegGLP-1 significantly reduced the number of apoptotic CD31^+^ cells compared to the solvent culture, while we did not observe gender differences in the severity of the drug effects (Figure 9g).

GLP-1 and pegGLP-1 significantly increased the expression of CD34 marker in CD31^+^ culture of lung cells in all the studied f1 and m1 and f4 and m4 groups (Figure 9a,c,f). We did not observe gender differences in the severity of the drug effects.

GLP-1 and pegGLP-1 increased the number of CD31^+^ lung cells with active esterases. This effect was more pronounced in the f1 and m1 groups than in the f4 and m4 groups (Figure 9b,e).

## 3. Discussion

MetS is a complex clinical condition, and abdominal visceral obesity is considered to be one of its major components [21]. Gender differences in the development and progression of obesity and MetS, as well as their treatment, represent a current challenge in clinical practice. Traditionally, gender differences in metabolism (mechanisms of accumulation of adipose tissue and mobilization of fat reserves, homeostasis of glucose, secretion, and effects of insulin) and related diseases (MetS and type 1 and type 2 diabetes) are explained by the effects of sex hormones [2,7]. In our study, we focused on the combination of obesity (or MD) and COPD. Obesity contributes to increased respiratory reactivity and can lead to various respiratory pathologies, including COPD, asthma, and other lung diseases [17,22]. Moreover, obesity is common in patients diagnosed with COPD and contributes to respiratory symptoms [23]. Importantly, endothelial dysfunction and endothelial disorders are important risk factors for many complications of COPD.

While studying healthy C57BL/6 mice, we found fewer EPC (CD45^−^CD31^+^CD34^+^; CD31^+^CD34^+^CD146^+^) and angiogenesis precursors (CD45^−^CD309^+^CD117^+^) in the lungs and bone marrow of females compared to males (Figure 6 and Figure 7). In vitro, no differences between male and female mice were found regarding the numbers of apoptotic cells, expression of CD34, and the number of cells with active esterase (Figure 9).

Administration of MSG induced dyslipidemia and obesity in both male and female animals (Figure 2). The changes in lipid metabolism in females of group f2 were less pronounced compared to males in group m2. Our data is in general accordance with previously published results [24]. As in our earlier study [20], this work revealed infiltration of the parenchyma of lungs by inflammatory cells (predominantly macrophages and single neutrophils) in mice of both sexes treated with MSG (Figure 4). At the same time, hemodynamic disturbances and a decrease in the expression of CD31 in the lungs were observed. In female animals in group f2, recruitment of bone marrow EPC, angiogenic precursor cells, circulating pericytes, and vascular smooth muscle cells into the lungs was found (Figure 8). It has been reported that pericytes and vascular smooth muscle cells are involved in the restoration of the normal structure and function of the damaged endothelium through intercellular contacts [25]. In accordance with this report, the recruitment of these cell types into the lungs could be explained by ongoing regeneration of the damaged endothelium in the lungs of MSG-treated females.

We also found more pronounced hemodynamic disturbances and low CD31 expression in the lungs of male animals within the m2 group compared to females in the f2 group (Figure 5). Surprisingly, this damage to the pulmonary endothelium in male animals did not lead to the recruitment of bone marrow EPC and angiogenic precursor cells into the lung tissue. Additionally, animals in group m2 showed signs of hyperglycemia and had an increased ratio of TG/HDL (Figure 2e). High values of the TG/HDL ratio and glucose tolerance are thought to indicate a high risk of vascular complications [26]. This suggests that TG/HDL and hyperglycemia could have a prognostic significance in the simulation of emphysema in mice with obesity. LPS and CSE introduction increased the inflammatory response and caused the formation of emphysema in mice of both sexes with obesity. Meanwhile, damage to the microvascular bed in the lungs was more profound in males in the m4 group (high values of TG/HDL and hyperglycemia before emphysema modeling) than in females of the f4 group (low values of TG/HDL and the normal level of glucose before emphysema modeling) (Figure 2e and Figure 4).

As evidenced by the findings above, the differences in the reaction of lung endothelium of females and males to external factors correlate with gender differences in fat metabolism and glucose metabolism.

LPS and CSE led to a decrease in the numbers of EPC (CD45^−^CD31^+^CD34^+^; CD31^+^CD34^+^CD146^+^), angiogenic precursor cells, vascular smooth muscle cells, cells of the lumen of the nascent vessels, and pericytes in the lungs of male mice in the group m4. These flow cytometric data can be explained by an impaired mobilization and migration of the cells. In f4 females, we observed recruitment of CD45^−^CD31^+^CD34^+^ EPC, vascular smooth muscle cells, and pericytes to the lungs (Figure 6). At the same time, mobilization and migration of angiogenic precursor cells in females in the f4 group, as well as in males in group m4, were disturbed. An analysis of CD31^+^ cells isolated from females in group f4 and males in group m4 in vitro revealed that there were no sex-specific differences in the rate of apoptosis, expression of CD34, and activity of esterase (Figure 9).

The main role of adult stem cells (SC) is the formation of new cells after injury [27]. In tissues of an adult organism, SCs are contained in the bone marrow and tissue-specific niches [28]. Markers of immature endothelial cells have been detected in bone marrow [28]. SC activity is regulated by internal mechanisms and external signals; the latter can come from a niche. It is believed that inflammation changes many homeostatic parameters and, thus, has a strong effect on various cells, including SC [29]. A negative impact of inflammation on the stem cell niche has been reported for intestinal stem cells, satellite cells or myogenic precursors cells, hepatic progenitor cells, epidermal stem cells, and neural stem cells [29]. Additionally, the mobilization of mesenchymal stromal/stem cells (MSC) by inflammatory factors has been demonstrated [30,31,32]. In our study, we found high levels of inflammatory cytokines in the lungs and disruption of EPC mobilization into the bloodstream in male mice in the m4 group and females in the group f4. In light of these findings, it is likely that mobilization of bone marrow EPC and MSC into the bloodstream and their migration to the damaged tissue is regulated by different mechanisms.

Therapy of obesity and its complications is currently limited by the lack of consideration of gender differences. Decreasing hyperglycemia and obesity could facilitate reducing the risk of vascular complications and related diseases [33]. It is well known that GLP-1 stimulates insulin production by islet β-cells, counteracts insulin resistance, improves peripheral glucose tolerance, and has anti-inflammatory properties [34,35]. In addition to the endocrine activity of GLP-1, it may also play a role in the homeostasis of the lungs. GLP-1 receptors are abundant in the alveoli, septum, airway, and smooth muscle of pulmonary vessels [36,37,38]. Moreover, their levels are relatively higher in the lungs than in the intestines and brain [39].

Known pegylated hormone analogs are characterized by improved pharmacokinetic characteristics without reducing the effectiveness of treatment and safety compared with native GLP-1 [40]. In our previous study, we assessed gender differences in the effects of pegGLP-1 in a streptozotocin-induced model of diabetes. Briefly, pegGLP-1 showed an anti-diabetic effect [41,42]. In the present study, pegGLP-1 showed more pronounced positive effects on the AUC, the ratio of TG/HDL, and emphysema square-extended alveolar tissue in females in group f6 compared to the f5 females treated with unpegylated GLP-1 (Figure 2). In addition, in female mice in f5 and f6 groups, both treatment regimes (pegGLP-1 and GLP-1) increased expression of CD31 in alveolar tissue. We attributed to additional recruitment of EPC (CD45^−^CD31^+^CD34^+^) and angiogenic precursors to the lungs. These cellular effects were more pronounced in the case of pegGLP-1 than GLP-1. We found no differences in the effect of both agents on cultivated CD31^+^ lung cells. Administration of pegGLP-1 or GLP-1 in cell culture resulted in a decrease of CD31^+^ endothelial cell apoptosis and an increase in the number of CD34+ cells with active esterases (Figure 9). Thus, it is possible that similar to native GLP-1 [20], CD31 and CD34 positive endothelial progenitor cells are the target for pegGLP-1.

In the present study, we also assessed potential gender differences in the effects of GLP-1 and pegGLP-1. GLP-1 and pegGLP-1 administration in the m5 and m6 groups significantly reduced serum triglyceride levels, BMI, and AUC compared to groups f5 and f6, whereas increased serum HDL cholesterol concentration in male animals (Figure 1, Figure 2, Figure 3). Both GLP-1 and pegGLP1 administration did not affect the area of emphysema in groups m5 and m6, while the expression of CD31 in the lungs increased. However, this increase in CD31 expression was not as strong as in female groups (Figure 5). Unlike in female animals in f5 and f6 groups, angiogenic precursor cells, vascular smooth muscle cells (by pegGLP-1), and pericytes (by GLP-1) were recruited into the damaged alveolar tissue of males of groups m5 and m6 (Figure 6).

In sum, we presented evidence of gender-specific differences in lung injury, mobilization and migration of EPC, and angiogenic precursor cells in mice with MD and lung emphysema. From our point of view, genetic factors controlling fat metabolism and glucose metabolism might be involved in the gender-specific differences, and these will be a subject of future studies. In addition, our data indicated differences in the effectiveness of pegGLP-1 in COPD/MetS comorbidity. These results also suggested potentially higher therapeutic effects of pegGLP-1 for COPD treatment in obese women (or women with MD), including older women. Finally, we proposed that CD31 and CD34 positive EPCs were the cellular targets of pegGLP-1.

## 4. Materials and Methods

### 4.1. Animals

Experiments were carried out on female and male C57BL/6 mice (certified animals from the nursery of E.D. Goldberg Research Institute of Pharmacology and Regenerative Medicine) in strict adherence to the principles of European Convention for the Protection of Vertebrate Animals used for Experimental and other Scientific Purposes (Strasbourg, 1986). The study was approved by the Institutional Animal Care and Use Committee (IACUC) of the E.D. Goldberg Research Institute of Pharmacology and Regenerative Medicine (license number IACUC No. 114062016, 22.06.2016). The day of birth was considered as experimental day 0 (p0).

### 4.2. Induction of Obesity

Female and male C57BL/6 mice received a daily subcutaneous (sc) injection of monosodium glutamate (MSG; Sigma, St. Louis, MO, USA) diluted in buffer solution (physiological saline) at a dose of 2.2 mg/g from p0 to p10 [42]. Physiological saline was injected into control mice in equivalent volume. Obesity parameters were estimated according to the Lee index on p124th [42,43,44]. Briefly, Lee index was calculated as a cubic root of body weight (g) × 10/nasoanal length (mm), where an index equal to or lower than 0.300 was classified as normal. Female and male mice with Lee index values higher than 0.300 were classified as obese and included in the study [45].

### 4.3. Exposure to Cigarette Smoke Extract

Cigarette smoke extract (CSE) was generated from L&M RED LABEL cigarettes (2 cigarettes/mL). The composition of the cigarettes was as follows: resin 10 mg/Cigarette, nicotine 0.8 mg/ Cigarette, CO 10 mg/CIG. Before obtaining the extract, the cigarette filter was removed; the length of a cigarette with the filter was 80 mm, 55 mm with the removed filter. The extraction was carried out by stretching the smoke of a lit cigarette through the phosphate buffer at a constant speed with the help of a vacuum pump; the cigarette was burned to a length of 5 mm. The burning time of one cigarette was 180 s. To remove the particles, the extract was filtered through a bacterial filter with a pore size of 45 nm. To standardize the obtained extract, pH (pH~7) and optical density were measured at wavelengths of 405 and 540 nm (D405~237, D540~123) before and after filtration.

On p126, lung emphysema was induced by intratracheal administration of lipopolysaccharide (LPS, Sigma, St. Louis, MO, USA) and CSE [46,47]. LPS at a dose of 3 μg/mouse in 50 μL phosphate buffer and 50 μL CSE were administered intratracheally. For the introduction of LPS and CSE, general anesthesia (pentobarbital) was used. LPS was administered on p126 and p129. CSE was introduced on p127, p130, 133, p136, p139, p142, p149, p156, p163, and p170 (Figure 10).

### 4.4. Pharmacological Compounds

Glucagon-like peptide-1 (GLP-1) was obtained from Sigma (St. Louis, MO, USA). Pegylation on free amino groups of the peptide was carried out using succinimide pegylating agent Sunbright ME-120 TS (NOF America Corporation, San Mateo, CA, USA), for which 1 mg of lyophilizate was dissolved in 2 mL of 20 mm Na-phosphate buffer, pH 7, containing 0.01% TWEEN 20 (Sigma, St. Louis, MO, USA), then 20 mg of pegylating agent (NOF America Corporation, San Mateo, CA, USA) was added to the solution. The reaction was stopped by the application of 200 µL of 0.1 M glycine solution. The molecular weight of GLP-1 was assessed by electrophoresis in polyacrylamide gel with SDS (Sodium dodecyl sulfate, Sigma, St. Louis, MO, USA) using a standard technique. Prior to pegylation, the samples contained 100% GLP-1. After pegylation, the samples contained 7% GLP- 1, 78% of monopeg-GLP-1, and 15% double peg-GLP-1.

GLP-1 and pegGLP-1 were daily administered intraperitoneally in the region of the pancreas at a dose of 3 mmol/kg on p149, 156, 157, 173, 184, 186, and 188 (Figure 10).

### 4.5. Experimental Groups

Healthy mice treated with a saline solution formed the control groups: female control group (f1), and male control group (m1) (Table 1). Mice with metabolic disorders (MD) were divided into two groups: metabolic disorders group, females (f2), metabolic disorders group, males (m2). Mice with lung emphysema were divided into two groups: lung emphysema group, females (f3), lung emphysema group, males (m3). Mice with metabolic disorders and pulmonary emphysema were divided into two groups: metabolic disorders and pulmonary emphysema group, females (f4), and metabolic disorders and pulmonary emphysema, males (m4). GLP-1-treated mice with metabolic disorders and emphysema of the lungs were divided into two groups: GLP-1 treatment of metabolic disorders and emphysema, females (f5), and GLP-1 treatment of metabolic disorders and emphysema, males (m5). PegGLP-1-treated mice with metabolic disorders and emphysema of the lungs were divided into two groups: pegGLP-1 treatment of metabolic disorders and emphysema group, females (f6), and pegGLP-1 treatment of metabolic disorders and lung emphysema group, males (m6). All mice were culled on p189 by CO_2_ asphyxia.

### 4.6. Body Mass Index (BMI)

On the day before culling (p188), BMI and Lee indexes were calculated. Briefly, BMI was calculated using the formula:
BMI = body weight (g)/body length^2^ (cm)
where body length was measured from the tip of the nose to the anus [43]. The Lee index was calculated as described above.

### 4.7. Glucose Tolerance Test (GTT)

Blood glucose levels were measured using a glucometer (Accu-Chek Performa Nano (Roche Diagnostics GmbH, Mannheim, Germany). A glucose tolerance test was performed on p188. Measurements of the initial level of glucose in the blood of animals were performed after 12 h of food deprivation. Subsequently, glucose was administered intragastrically (D-glucose, Sigma, St. Louis, MO, USA) at a dose of 2 g/kg. Blood samples to study glucose levels were taken 0, 15, 30, 60, 90 min after glucose administration [48].

### 4.8. Lipid Profile Determination

The lipid profile was determined on p189. Blood samples were taken from each animal in tubes without additives, kept at room temperature for 30 min, and then centrifuged at 300× *g* for 30 min. The serum was separated and used to study lipid profile parameters. The concentration of cholesterol and TG was determined by direct enzymatic methods using BioSystems reagents (Barcelona, Spain) in accordance with the manufacturer’s instructions. Fractions of cholesterol, TG, high-density lipoproteins (HDL), low-density lipoproteins (LDL), and very-low-density lipoproteins (VLDL) were precipitated with phosphor-wolframate and polyvinyl sulfate, respectively, and their concentration was determined by the level of residual cholesterol. All the results were expressed as mmol/l. On p189, TG/HDL was assessed, as previously described in [49,50].

### 4.9. Lung Tissue Histology

The morphological examination of lungs was performed on p189. Briefly, the left lobe of the lung was fixed in a 10% solution of neutral formalin, carried out through alcohols of ascending concentrations to xylene, and poured into paraffin according to a standard procedure. Five micrometer thick dewaxed cuts were stained with hematoxylin and eosin [51]. Micro-preparations from each experimental animal were examined an Axio Lab.A1 light microscope (Carl Zeiss, MicroImaging GmbH, Göttingen, Germany) at 100× and 400× magnifications. Histoarchitecture of lung tissue and pathophysiological features of the tissue, including the presence of edema and inflammatory infiltration, venous congestion, as well as thickening of vessel walls and bronchi, were assessed [48,52,53].

### 4.10. Flow Cytometry

Mononuclear cells from the blood, bone marrow, and lung tissue were obtained, as previously described on p189 [20,54], followed by flow cytometric analysis of the expression of surface markers of mouse mononuclear cells. Briefly, cell suspensions were stained with the following fluorophore-conjugated monoclonal antibodies: CD45 PerCP, CD31 APC, CD34 FITC, CD146 PerCP-Cy5.5, CD309 (Flk-1) APC, and CD117 (c-kit) PeCy7 (all Becton Dickinson, San Jose, CA, USA). Appropriate isotype controls were used. Labeled cells were thoroughly washed with PBS and analyzed on a FACSCanto II flow cytometer (Becton Dickinson, San Jose, CA, USA) using FACS Diva software. At least 100,000 events were recorded for each sample.

### 4.11. Lung Tissue Dissociation and Magnetic Separation of CD31^+^ Cells

The effects of GLP-1 and pegGLP-1 on CD31^+^ lung cells in vitro were studied on p189 in cells isolated from an animal in the groups f1, m1, f4, and m4. Lungs were isolated, and the lung tissue was mechanically and enzymatically dissociated, followed by magnetic sorting for CD31^+^, as previously described [20].

### 4.12. Cultivation of CD31^+^ Cells

After 5-days of cultivation, CD31^+^ cells from f1, m1, f4, and m4 mice were harvested using tryptic digestion, and the cells were plated in a concentration of 3 × 10^5^ cells/1 mL medium in gelatin-coated flasks. M199 standard cultivation medium was supplemented with GLP-1 (10^−7^ M) or pegGLP-1 (10^−7^ M) (Table 2) followed by cultivation under standard conditions (3.5% CO_2_, 37 °C) for 24 h. In the following, the effects of GLP-1 and pegGLP-1 on CD31^+^ cells were evaluated by flow cytometry and imaging using the Cytation™ 3 imaging system [20].

### 4.13. Cellular Imaging

Images of CD31^+^ cells were obtained using a Cytation 3 Cell Imaging multimode reader (BioTek Instruments, Inc., Winooski, VT, USA) equipped with DAPI, GFP, and Texas Red light cubes. Cells were stained with Hoechst 33342, Annexin V-iFluor ™ 350 CFSE, and 7-AAD. Images were analyzed using Gen5 ™ data analysis software (Bad Friedrichshall, Germany), as described before [20].

### 4.14. Statistical Analysis

Statistical analysis was performed using SPSS statistical software (version 15.0, SPSS Inc., Chicago, IL, USA). Data were analyzed and presented as means ± standard error of the mean. Statistical significance was evaluated by Student’s *t*-test (for parametric data), or Mann–Whitney test (for nonparametric data) was used according to distribution. A *p*-value of less than 0.05 (by two-tailed testing) was considered an indicator of statistical significance.

## Figures and Tables

**Figure 1 ijms-20-05414-f001:**
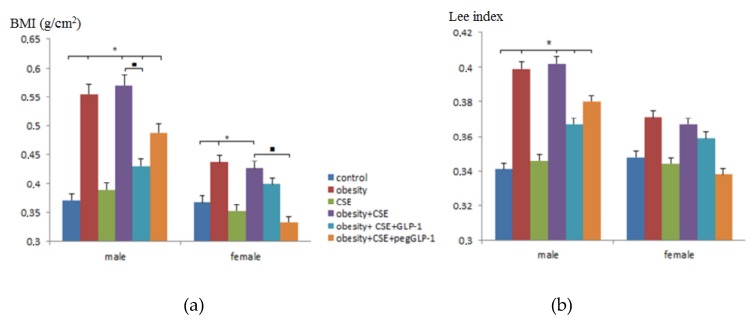
The effect of glucagon-like peptide-1 (GLP-1) and pegylated GLP-1 (pegGLP-1) on body mass index (BMI) and Lee index of male and female C57BL/6 mice on p189: (**a**) The BMI (g/cm^2^); (**b**) The Lee index. Groups: control—a control group from intact mice, obesity—mice with metabolic disorders (obesity and hyperglycemia), CSE—mice with lungs emphysema, obesity+CSE—mice with metabolic disorders (obesity and hyperglycemia) and lungs emphysema, obesity+CSE+GLP-1—mice with metabolic disorders (obesity and hyperglycemia) and lungs emphysema treated with GLP-1, obesity+CSE+pegGLP-1—mice with metabolic disorders (obesity and hyperglycemia) and lungs emphysema treated with pegGLP-1. Results are presented as the mean±SEM. *—significance of difference compared with control (*p* < 0.05); ■—significance of difference compared with the obesity+CSE group (*p* < 0.05). CSE, cigarette smoke extract.

**Figure 2 ijms-20-05414-f002:**
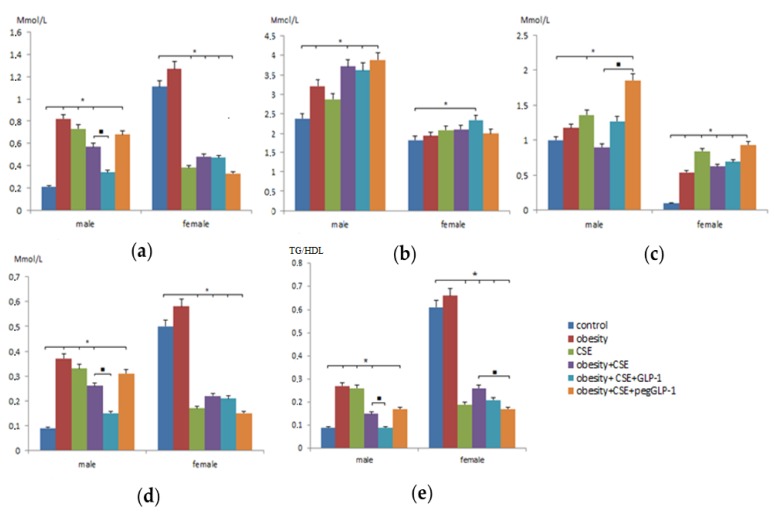
Lipid profile measurements in the blood of female and male C57BL/6 mice on p189: (**a**) The level of triglycerides in serum (Mmol/l); (**b**) High-density lipoprotein level (Mmol/l); (**c**) Low-density lipoprotein level (Mmol/l); (**d**) Very low-density lipoprotein level (Mmol/l); (**e**) The ratio of triglycerides to high-density lipoproteins (TG/HDL). Groups: control—a control group from intact mice, obesity—mice with metabolic disorders (obesity and hyperglycemia), CSE—mice with lungs emphysema, obesity+CSE—mice with metabolic disorders (obesity and hyperglycemia) and lungs emphysema, obesity+CSE+GLP-1—mice with metabolic disorders (obesity and hyperglycemia) and lungs emphysema treated with GLP-1, obesity+CSE+pegGLP-1—mice with metabolic disorders (obesity and hyperglycemia) and lungs emphysema treated with pegGLP-1. *—significance of difference compared with control (*p* < 0.05); ■—significance of difference compared with the obesity+CSE group (*p* < 0.05).

**Figure 3 ijms-20-05414-f003:**
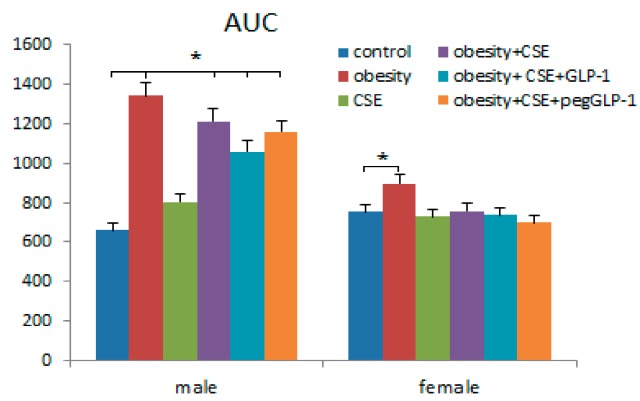
The area under the curve (AUC) of female and male C57BL/6 mice during the glucose tolerance test (on p188). Groups: control—a control group from intact mice, obesity—mice with metabolic disorders (obesity and hyperglycemia), CSE-mice with lungs emphysema, obesity+CSE—mice with metabolic disorders (obesity and hyperglycemia) and lungs emphysema, obesity+CSE+GLP-1—mice with metabolic disorders (obesity and hyperglycemia) and lungs emphysema treated with GLP-1, obesity+CSE+pegGLP-1—mice with metabolic disorders (obesity and hyperglycemia) and lungs emphysema treated with pegGLP-1. *—significance of difference compared with control (*p* < 0.05).

**Figure 4 ijms-20-05414-f004:**
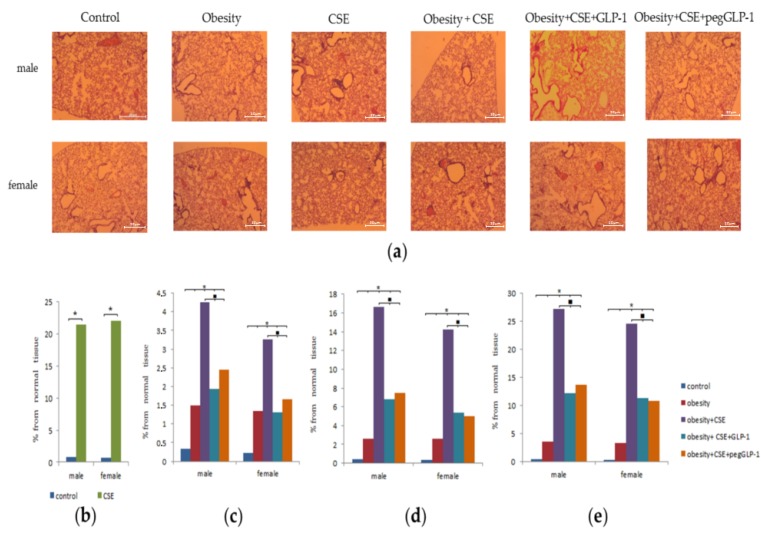
Morphological study of lung obtained from male and female C57BL/6 mice (*n* = 6): (**a**) Photomicrographs of left lung sections (lower pulmonary field) (on p189). Tissues were stained with hematoxylin-eosin; (**b**) The area of emphysema-expanded lung tissue (lower pulmonary field) of mice from all groups (on p148); (**c**) The area of emphysema-expanded lung tissue (upper pulmonary field) of mice from all groups (on p); (**d**) The area of emphysema-expanded lung tissue (middle pulmonary field) of mice from all groups (on p189); (**e**) The area of emphysema-expanded lung tissue (lower middle pulmonary field) of mice from all groups (on the p189). Groups: control—a control group from intact mice, obesity—mice with metabolic disorders (obesity and hyperglycemia), CSE—mice with lungs emphysema, obesity+CSE—mice with metabolic disorders (obesity and hyperglycemia) and lungs emphysema, obesity+CSE+GLP-1–mice with metabolic disorders (obesity and hyperglycemia) and lungs emphysema treated with GLP-1, obesity+CSE+pegGLP-1—mice with metabolic disorders (obesity and hyperglycemia) and lungs emphysema treated with pegGLP-1. * *p* < 0.05 significance of difference compared with control group, ■—significance of difference compared with the obesity+CSE group (*p* < 0.05).

**Figure 5 ijms-20-05414-f005:**
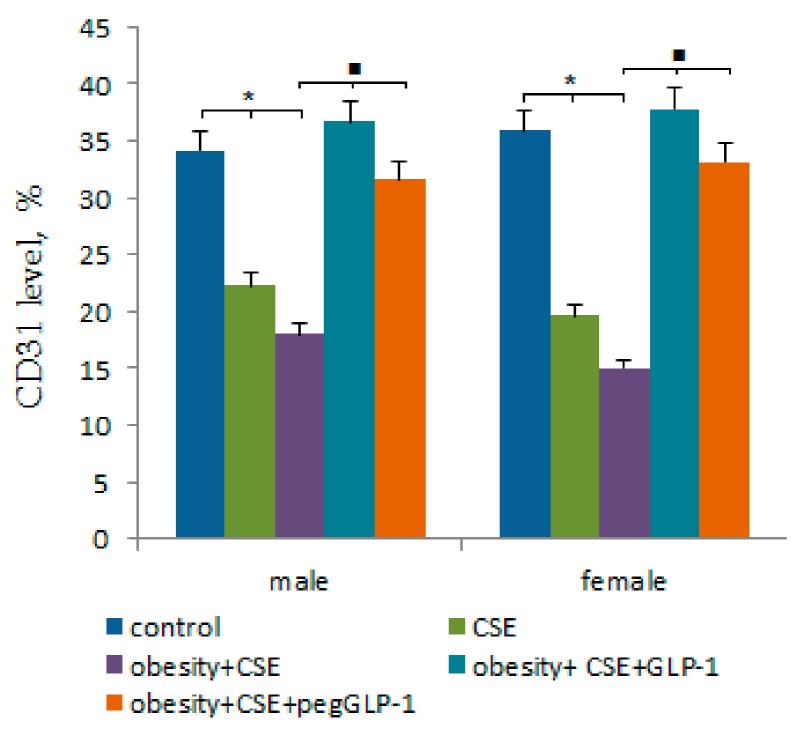
The relative content of cells expressing CD31 antigen in the lungs isolated from male and female C57BL/6 mice at the immunohistochemical staining for specific cellular marker: CD31 (on the p189). Groups: control—a control group from intact mice, obesity—mice with metabolic disorders (obesity and hyperglycemia), CSE—mice with lungs emphysema, obesity+CSE—mice with metabolic disorders (obesity and hyperglycemia) and lungs emphysema, obesity+CSE+GLP-1—mice with metabolic disorders (obesity and hyperglycemia) and lungs emphysema treated with GLP-1, obesity+CSE+pegGLP-1—mice with metabolic disorders (obesity and hyperglycemia) and lungs emphysema treated with pegGLP-1. **p* < 0.05 significance of difference compared with control group, ■—significance of difference compared with the obesity+CSE group (*p* < 0.05).

**Figure 6 ijms-20-05414-f006:**
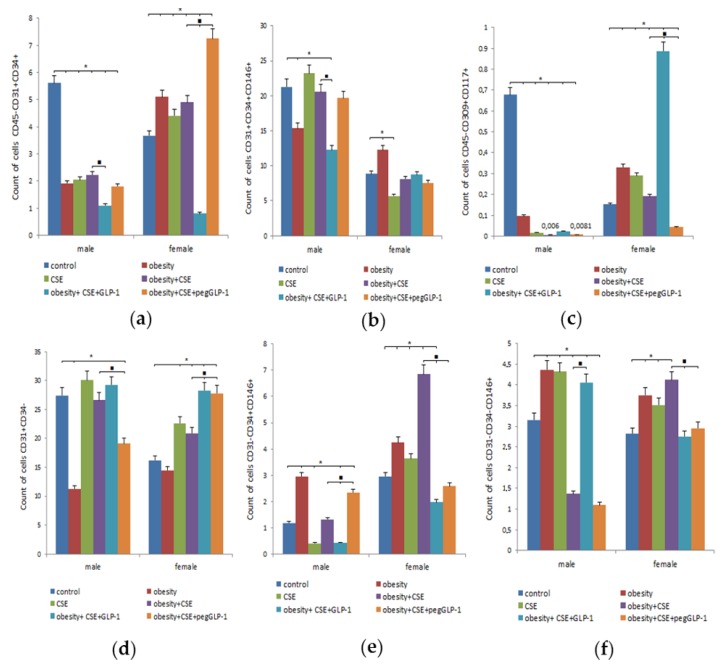
Characterization of cell population isolated from the lung of male and female C57BL/6 mice on the p189. (**a**) The content of endothelial progenitor cells (EPC) (CD45^−^CD31^+^CD34^+^); (**b**) The content of EPC (CD31^+^CD34^+^CD146^+^); (**c**) The content of angiogenesis precursors (CD45^−^CD309^+^CD117^+^); (**d**) The content of cells in the lumen of the nascent vessel (CD31^+^CD34^−^); (**e**) The content of vascular smooth muscle cells (CD31^−^CD34^+^CD146^+^); (**f**) The content of pericytes (CD31^−^CD34^−^CD146^+^). Cells were analyzed by flow cytometry using antibodies for CD31, CD34, CD45, CD146, CD117, CD309 mice. Dot plots are representative of three independent experiments with the mean from three independent experiments. Groups: control—a control group from intact mice, obesity—mice with metabolic disorders (MD) (obesity and hyperglycemia), CSE—mice with lungs emphysema, obesity+CSE—mice with MD (obesity and hyperglycemia) and lungs emphysema, obesity+CSE+GLP-1—mice with MD (obesity and hyperglycemia) and lungs emphysema treated with GLP-1, obesity+CSE+pegGLP-1—mice with MD (obesity and hyperglycemia) and lungs emphysema treated with pegGLP-1. *—significance of difference compared with control (*p* < 0.05); ■—significance of difference compared with the obesity+CSE group (*p* < 0.05).

**Figure 7 ijms-20-05414-f007:**
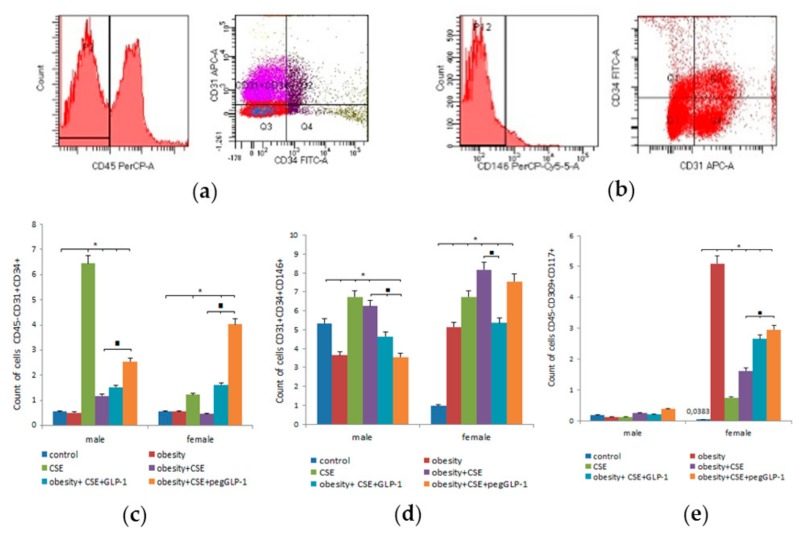
Characterization of cell population isolated from the bone marrow of male and female C57BL/6 mice on the p189. (**a**) Phenotype establishment and qualitative analysis of CD45 (PerCP), CD34 (FITC), and CD31 (APC) expression; (**b**) Phenotype establishment and qualitative analysis of CD34 (FITC), CD31 (APC), and CD146 (PerCP-Cy5.5) expression; (**c**) The content of EPC (CD45^−^CD31^+^CD34^+^); (**d**) The content of EPC (CD31^+^CD34^+^CD146^+^); (**e**) The content of angiogenesis precursors (CD45^−^CD309^+^CD117^+^). Cells were analyzed by flow cytometry using antibodies for CD45, CD31, CD34, CD146, CD117, CD309 mice. Dot plots are representative of three independent experiments with the mean from three independent experiments. Groups: control—a control group from intact mice, obesity—mice with MD (obesity and hyperglycemia), CSE—mice with lungs emphysema, obesity+CSE—mice with MD (obesity and hyperglycemia) and lungs emphysema, obesity+CSE+GLP-1—mice with MD (obesity and hyperglycemia) and lungs emphysema treated with GLP-1, obesity+CSE+pegGLP-1—mice with MD (obesity and hyperglycemia) and lungs emphysema treated with pegGLP-1. *—significance of difference compared with control (*p* < 0.05); ■—significance of difference compared with the obesity+CSE group (*p* < 0.05).

**Figure 8 ijms-20-05414-f008:**
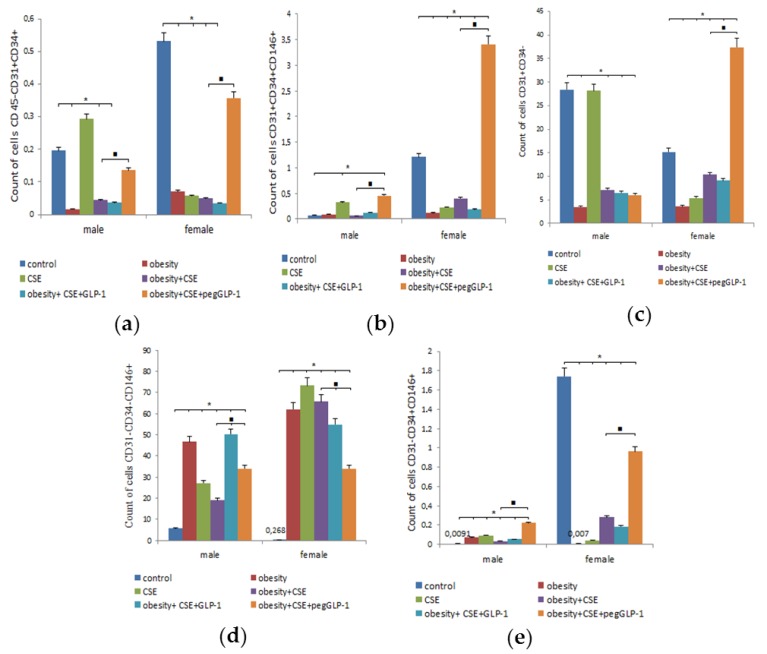
Characterization of cell population isolated from the blood of male and female C57BL/6 mice on the p189. (**a**) The content of EPC (CD45^−^CD31^+^CD34^+^); (**b**) The content of EPC (CD31^+^CD34^+^CD146^+^); (**c**) The content of cells in the lumen of the nascent vessel (CD31^+^CD34^−^); (**d**) The content of pericytes (CD31^−^CD34^−^CD146^+^); (**e**) The content of vascular smooth muscle cells (CD31^−^CD34^+^CD146^+^). Cells were analyzed by flow cytometry using antibodies for CD45, CD31, CD34, CD146 mice. Dot plots are representative of three independent experiments with the mean from three independent experiments. Groups: control—a control group from intact mice, obesity—mice with MD (obesity and hyperglycemia), CSE—mice with lungs emphysema, obesity+CSE—mice with MD (obesity and hyperglycemia) and lungs emphysema, obesity+CSE+GLP-1—mice with MD (obesity and hyperglycemia) and lungs emphysema treated with GLP-1, obesity+CSE+pegGLP-1—mice with MD (obesity and hyperglycemia) and lungs emphysema treated with pegGLP-1. *—significance of difference compared with control (*p* < 0.05); ■—significance of difference compared with the obesity+CSE group (*p* < 0.05).

**Figure 9 ijms-20-05414-f009:**
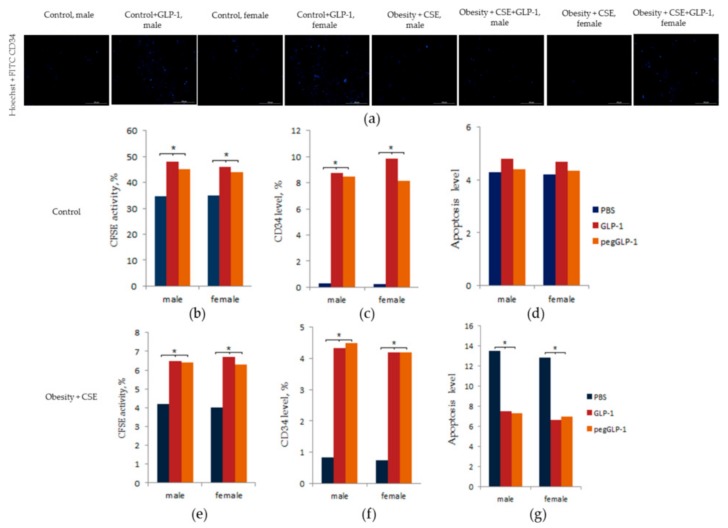
GLP-1 and pegGLP-1 treatment effects on CD31^+^ endothelial cells isolated from the lungs of male and female C57BL/6 mice in vitro: (**a**) Images of CD31^+^ cells stained with: Hoechst (blue) to identify cell nuclei; CD34 FITC (green) (Hoechst + CD34) composite image using all two colors. All scale bars are 100 µm; (**b**–**g**) CD31+ endothelial cells from lung were precultured for 5 days, incubated with or without GLP-1 (10^−7^ M) or pegGLP-1 (10^−7^ M) for 24 h and then labeled with Hoechst, Carboxyfluorescein succinimidyl ester (CFSE) (**b**,**e**) CD34 FITC (**c**,**f**), Annexin V and 7-Aminoactinomycin D (7-AAD) (**d**,**g**) prior to fluorescence microscopic analysis. (**b**) CFSE activity after culture of cells isolated from the lung of intact mice; (**c**) the level of CD34^+^ cells after culture of cells isolated from the lung of intact mice; (**d**) the count of cells with apoptosis after culture of cells isolated from the lung of intact mice; (**e**) CFSE activity after culture of cells isolated from the lung of mice with MD and lung emphysema; (**f**) the level of CD34^+^ cells after culture of cells isolated from the lung of mice with MD and lung emphysema; (**g**) the count of cells with apoptosis after culture of cells isolated from the lung of mice with MD and lung emphysema. All data are expressed as mean ± SD, *—significance of difference compared with control (*p* < 0.05).

**Figure 10 ijms-20-05414-f010:**
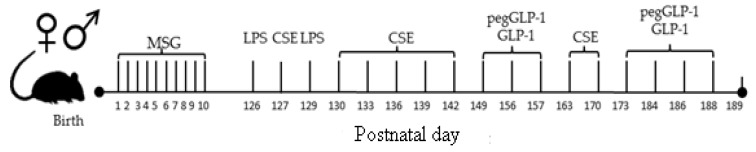
Schematic diagram of the experimental procedures.

**Table 1 ijms-20-05414-t001:** Experimental groups in vivo.

	Control Groups	Metabolic Disorders	Lung Emphysema	Metabolic Disorders + Lung Emphysema	Metabolic Disorders + Lung Emphysema + GLP-1	Metabolic Disorders + Lung Emphysema + pegGLP-1
Females	f1 ^1^ (*n* = 10)	f2 (*n* = 10)	f3 (*n* = 10)	f4 (*n* = 10)	f5 (*n* = 10)	f6 (*n* = 10)
Males	m1 ^2^ (*n* = 10)	m2 (*n* = 10)	m3 (*n* = 10)	m4 (*n* = 10)	m5 (*n* = 10)	m6 (*n* = 10)

^1^ f—Females, ^2^ m—males.

**Table 2 ijms-20-05414-t002:** Experimental groups in vitro.

Drugs Introduced into the Culture CD31^+^ Cells	CD31^+^ Cell Culture			
	Group f1 (Control Groups)	Group m1 (Control Groups)	Group f4 (MD + Lung Emphysema)	Group m4 (MD + Lung Emphysema)
PBS	+	+	+	+
GLP-1	+	+	+	+
pegGLP-1	+	+	+	+

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
