# Peer review of "Gender Differences in the Pharmacological Actions of Pegylated Glucagon-Like Peptide-1 on Endothelial Progenitor Cells and Angiogenic Precursor Cells in a Combination of Metabolic Disorders and Lung Emphysema"

_ijms, 2019, doi:10.3390/ijms20215414_

Round 1
Reviewer 1 Report
Topic is of interest and the study is large and contains relevant information about gender-specific mechanisms of COPD with MetS and significant contribution. The work has a good analysis. There are appropriate and adequate reference.
Considering that one of the important result of study was the differences in the reaction of lung endothelium of females and males to external factors correlate with gender differences of fat metabolism and glucose metabolism. However, in discussion this results do not contextualization. Suggests further developing the discussion and contestualizing the results with data related to genetic factors.
Author Response
To the Reviewer #1
International Journal of Molecular Sciences
Dear Reviewer#1
Thank you for your review. We are very grateful to the reviewer for your work in reviewing our manuscript. We revised the manuscript. Below please find the reviewers’ comments and our responses. The elucidations and the changes have been included into the revised version.
Author's Responses to Questions
Point 1: Topic is of interest and the study is large and contains relevant information about gender-specific mechanisms of COPD with MetS and significant contribution. The work has a good analysis. There are appropriate and adequate reference.
Considering that one of the important result of study was the differences in the reaction of lung endothelium of females and males to external factors correlate with gender differences of fat metabolism and glucose metabolism. However, in discussion this results do not contextualization. Suggests further developing the discussion and contestualizing the results with data related to genetic factors.
Response 1: We corrected all the text of our manuscript.
A link between metabolic syndrome (MetS) or better to metabolic disorder (MetD), and lung diseases has been reported by several cross-sectional and longitudinal studies. There are reports about gender-specific both MetS and COPD. However, the mechanism of gender difference is not clear. The findings may uncover mechanisms for the difference in clinical course in female COPD and MetS patients compared to male patients. The findings of our paper highlight a potential gender disparity in the development and treatment of COPD and MetS.
The goal of our research was to study gender differences in the treatment of hyperglycemia, dyslipidemia and obesity, emphysema, and alveolar endothelial injury with pegylated GLP-1 (pegGLP-1) in a C57BL/6 mice model of obesity and emphysema.
Reviewer 2 Report
This is an extensive and interesting study on pathophysiology and alterations of pulmonary tissue (emphysema, COPD). There are certain constraints in reading and understanding the study:
The results are presented prior to the investigated material and methods. This unusual order should be changed. No human data are given (at least from the literature) of morphology differences in the lungs prior to sexual maturation (for example see: Kayser, Height and Weight in Human Beings, VaW, Munich, ISBN: 3-922251-99-4. Herein (in agreement with the findings of the authors) no differences in gross appearance (size and weight of autopsied girls and boys) could be documented. A reproducible morphometry of lung is difficult. At least some techniques should be mentioned. For example. stereology (Weigel et al), diffusion (structure - function relationship) KAYSER, Klaus; BORKENFELD, Stephan; KAYSER, Gian. Digital Image Content and Context Information in Tissue-based Diagnosis. Diagnostic Pathology, [S.l.], v. 4, n. 1, dec. 2018. ISSN 2364-4893), DEROULERS, Christophe et al. Automatic quantification of the microvascular density on whole slide images, applied to paediatric brain tumours. Diagnostic Pathology, [S.l.], v. 2, n. 1, sep. 2016. ISSN 2364-4893. Available at: <http://www.diagnosticpathology.eu/content/index.php/dpath/article/view/209>. It remains unclear how the emphysema, edema, etc. of the lungs was defined and distinguished from (still normal) appearance. In addition, how was the grading of the alterations? In contrast to the quite extensive cellular and biochemical investigations, the analysis of the lung morphology seems to be quite poor. However, emphysema is a primarily a structure change, which can microscopically be measured by appropriate techniques (KAYSER, Gian et al. The application of structural entropy in tissue based diagnosis. Diagnostic Pathology, [S.l.], v. 3, n. 1, aug. 2017. ISSN 2364-4893. Available at: <http://www.diagnosticpathology.eu/content/index.php/dpath/article/view/251>. Date accessed: 01 oct. 2019. doi: https://doi.org/10.17629/www.diagnosticpathology.eu-2017-3:251.). The study would increase in interest if at least these options are briefly mentioned (and cited). A few characteristic microscopic images should be included to explain / visualize the underlying tissue structures. 7. The findings should be graded in order to their significance (contribution to answer the goal of the study)Author Response
To the Reviewer #2
International Journal of Molecular Sciences
Dear Reviewer#2
Thank you for your review. We are very grateful to the reviewer for your work in reviewing our manuscript. We revised the manuscript. Below please find the reviewers’ comments and our responses. The elucidations and the changes have been included into the revised version.
Author's Responses to Questions
Point 1: This is an extensive and interesting study on pathophysiology and alterations of pulmonary tissue (emphysema, COPD). There are certain constraints in reading and understanding the study:
The results are presented prior to the investigated material and methods. This unusual order should be changed. No human data are given (at least from the literature) of morphology differences in the lungs prior to sexual maturation (for example see: Kayser, Height and Weight in Human Beings, VaW, Munich, ISBN: 3-922251-99-4. Herein (in agreement with the findings of the authors) no differences in gross appearance (size and weight of autopsied girls and boys) could be documented. A reproducible morphometry of lung is difficult. At least some techniques should be mentioned. For example. stereology (Weigel et al), diffusion (structure - function relationship) KAYSER, Klaus; BORKENFELD, Stephan; KAYSER, Gian. Digital Image Content and Context Information in Tissue-based Diagnosis. Diagnostic Pathology, [S.l.], v. 4, n. 1, dec. 2018. ISSN 2364-4893), DEROULERS, Christophe et al. Automatic quantification of the microvascular density on whole slide images, applied to paediatric brain tumours. Diagnostic Pathology, [S.l.], v. 2, n. 1, sep. 2016. ISSN 2364-4893. Available at: <http://www.diagnosticpathology.eu/content/index.php/dpath/article/view/209>. It remains unclear how the emphysema, edema, etc. of the lungs was defined and distinguished from (still normal) appearance. In addition, how was the grading of the alterations? In contrast to the quite extensive cellular and biochemical investigations, the analysis of the lung morphology seems to be quite poor. However, emphysema is a primarily a structure change, which can microscopically be measured by appropriate techniques (KAYSER, Gian et al. The application of structural entropy in tissue based diagnosis. Diagnostic Pathology, [S.l.], v. 3, n. 1, aug. 2017. ISSN 2364-4893. Available at: <http://www.diagnosticpathology.eu/content/index.php/dpath/article/view/251>. Date accessed: 01 oct. 2019. doi: https://doi.org/10.17629/www.diagnosticpathology.eu-2017-3:251.). The study would increase in interest if at least these options are briefly mentioned (and cited). A few characteristic microscopic images should be included to explain / visualize the underlying tissue structures. 7. The findings should be graded in order to their significance (contribution to answer the goal of the study)
Response 1: We corrected all the text of our manuscript.
We presented manuscript according to template.
We investigated a link between metabolic syndrome (MetS) and chronic obstructive pulmonary disease (COPD). It is known that both as metabolic syndrome and as COPD has a gender differences. Increasingly, clinicians are recognizing that both as COPD and as MetS are heterogeneous diseases and there are many different phenotypes that comprise the diagnosis. While several studies have shown gender differences in therapy for other chronic diseases, there is limited research on how COPD and MetS management differs between men and women. The goal of our research was to study gender differences in the treatment of hyperglycemia, dyslipidemia and obesity, emphysema, and alveolar endothelial injury with pegylated GLP-1 (pegGLP-1) in a C57BL/6 mice model of obesity and emphysema. We investigated the effect of pegGLP-1 on bone marrow, circulating in the blood and pulmonary endothelial progenitor cells and other cells involved in angiogenesis in female and male animals to assess potential gender-specific differences.
We investigated the lung morphology by standard methods according to the literature data [1-4]. The hematoxylin and eosin stain is the standard used for microscopic examination of tissues that have been fixed, processed, embedded, and sectioned.
Damasceno, D.C.; Sinzato, Y.K.; Bueno, A.; Dallaqua, B.; Lima, P.H.; Calderon , I.M.P.; Rudge, M.V.C.; Campos, K.E. Metabolic Profile and Genotoxicity in Obese Rats Exposed to Cigarette Smoke. Obesity (Silver Spring). 2013, 21, 1569-1601. doi: 10.1002/oby.20152.· Parameswaran, H.; Majumdar, A.; Ito, S.; Alencar, A.M.; Suki, B. Quantitative characterization of airspace enlargement in emphysema. Appl. Physiol. 2006, 100, 186–193. doi: 10.1152/japplphysiol.00424.2005. He, Z.H.; Chen, P.; Chen, Y.; He, S.D.; Ye, J.R.; Zhang, H.L.; Cao, J. Comparison between cigarette smoke-induced emphysema and cigarette smoke extract-induced emphysema. Induc. Dis. 2015, 13, 6. https://doi.org/10.1186/s12971-015-0033-z. Munoz-Barrutia, A.; Ceresa, M.; Artaechevarria, X.; Montuenga, L.M.; Ortiz-de-Solorzano, C. Quantification of lung damage in an elastase-induced mouse model of emphysema. J. Biomed. Imaging 2012, 2012, 734734. http://dx.doi.org/10.1155/2012/734734.
There are human data of morphology differences in the lungs in the References of our manuscript:
Åberg, J.; Hasselgren, M.; Montgomery, S.; Lisspers, K.; Ställberg, B.; Janson, C.; Sundh, J. Sex-related differences in management of Swedish patients with a clinical diagnosis of chronic obstructive pulmonary disease. Int J Chron Obstruct Pulmon Dis. 2019, 14, 961-969. doi: 10.2147/COPD.S193311. Kim, J.H.; Yoo, J.Y.; Kim, H.S. Metabolic Syndrome in South Korean Patients with Chronic Obstructive Pulmonary Disease: A Focus on Gender Differences. Asian Nurs Res (Korean Soc Nurs Sci) 2019, 13, 137-146. doi: 10.1016/j.anr.2019.03.002. Chen, H.; Li, Z.; Dong, L.; Wu, Y.; Shen, H.; Chen, Z. Lipid metabolism in chronic obstructive pulmonary disease. International Journal of Chronic Obstructive Pulmonary Disease 2019, 14, 1009–1018. http://doi.org/10.2147/COPD.S196210. Zore, T.; Palafox, M.; Reue, K. Sex differences in obesity, lipid metabolism, and inflammation-A role for the sex chromosomes? Mol Metab. 2018, 15, 35-44. doi: 10.1016/j.molmet.2018.04.003. Lamonaca, P.; Prinzi, G.; Kisialiou, A.; Cardaci, V.; Fini, M.; Russo, P. Metabolic Disorder in Chronic Obstructive Pulmonary Disease (COPD) Patients: Towards a Personalized Approach Using Marine Drug Derivatives. Mar Drugs. 2017, 15, 81. doi: 10.3390/md15030081. García-Rio, F.; Soriano, J.B.; Miravitlles, M.; Muñoz, L.; Duran-Tauleria, E.; Sánchez, G.; Sobradillo, V.; Ancochea, J. Impact of obesity on the clinical profile of a population-based sample with chronic obstructive pulmonary disease. PLoS One. 2014, 9, doi: 10.1371/journal.pone.0105220.